# IoT Electrochemical Sensor with Integrated Ni(OH)_2_–Ni Nanowires for Detecting Formaldehyde in Tap Water

**DOI:** 10.3390/s23104676

**Published:** 2023-05-11

**Authors:** Špela Trafela, Abhilash Krishnamurthy, Kristina Žagar Soderžnik, Urška Kavčič, Igor Karlovits, Beno Klopčič, Sašo Šturm, Kristina Žužek

**Affiliations:** 1Department for Nanostructured Materials, Jožef Stefan Institute, Jamova c. 39, 1000 Ljubljana, Slovenia; abhilash.krishnamurthy@ijs.si (A.K.);; 2Jožef Stefan International Postgraduate School, Jamova c. 39, 1000 Ljubljana, Slovenia; 3Pulp and Paper Institute, Bogišićeva 8, 1000 Ljubljana, Slovenia

**Keywords:** formaldehyde, electrochemical sensor, nickel

## Abstract

Simple, low-cost methods for sensing volatile organic compounds that leave no trace and do not have a detrimental effect on the environment are able to protect communities from the impacts of contaminants in water supplies. This paper reports the development of a portable, autonomous, Internet of Things (IoT) electrochemical sensor for detecting formaldehyde in tap water. The sensor is assembled from electronics, i.e., a custom-designed sensor platform and developed HCHO detection system based on Ni(OH)_2_–Ni nanowires (NWs) and synthetic-paper-based, screen-printed electrodes (pSPEs). The sensor platform, consisting of the IoT technology, a Wi-Fi communication system, and a miniaturized potentiostat can be easily connected to the Ni(OH)_2_–Ni NWs and pSPEs via a three-terminal electrode. The custom-made sensor, which has a detection capability of 0.8 µM/24 ppb, was tested for an amperometric determination of the HCHO in deionized (DI) and tap-water-based alkaline electrolytes. This promising concept of an electrochemical IoT sensor that is easy to operate, rapid, and affordable (it is considerably cheaper than any lab-grade potentiostat) could lead to the straightforward detection of HCHO in tap water.

## 1. Introduction

Formaldehyde (HCHO) is a serious pollutant. It is classified as toxic, allergenic, and carcinogenic for humans, with a maximum permissible level of 0.1 ppm for homes and 1 ppm for workplaces [1,2]. It can be found in many everyday products, including disinfectants, cosmetics, textiles, and paint [1,2,3,4]. It can also be found in water, usually in a rapidly hydrated form (i.e., methylene glycol), and therefore it can easily enter drinking water as a by-product of disinfectants or as a leachate from plumbing fixtures.

Several electrochemical sensors have been developed to detect HCHO in the gas phase [2,5,6,7,8,9,10,11]. However, a cost-effective, miniaturized, and portable electrochemical sensor (lab-on-a-chip) to monitor aqueous solutions of HCHO (HCHO_aq_) is not available. In general, HCHO_aq_ is detected with expensive, bulky laboratory equipment, e.g., impractical electrochemical cells with a three-electrode system (working, counter, and reference electrodes) and large potentiostats with all the electronics required to control the cell and run the experiments. Although this equipment can measure HCHO_aq_ at the low-ppb level [2], its practical application for domestic use is prevented by cost, complexity, and a lack of portability [5]. These limitations can be overcome to provide portable, real-time monitoring by assembling miniaturized electronics, a three-terminal electrode connector, and Wi-Fi communication support [3,5,12,13] to provide an inexpensive, disposable, and portable detection system.

Furthermore, the practical advantage of electrochemical HCHO_aq_ detection could have future implications for low-cost, mass-produced devices using a three-electrode system based on screen-printed electrodes (SPEs) [14,15]. They are seen as ideal for developing portable, inexpensive sensors because of their simplicity, the flexibility of their surface modification, and their ease of disposability compared to traditional electrode materials [3]. Usually, commercially available SPEs, which are mainly printed on ceramics, are used for sensor-based detection systems. However, screen-printing technology offers the possibility to print conductive inks on lighter, ecological materials such as synthetic paper [16,17]. The paper substrates are mainly produced from chemical or mechanical pulp using a different ratio of fibers (e.g., hardwood and softwood) with additional components (e.g., fillers, retention aids, binders) that enhance the structural and surface properties of the paper [16]. Synthetic paper-based SPEs (pSPE) can also be modified with nanostructured materials to improve their electroanalytical characteristics for the detection of many harmful chemical species, including HCHO_aq_ [18,19].

In recent years, various types of nanostructured materials based on noble metals (Au [20,21], Pt [22], Ag [23], etc.) have been reported as promising for highly sensitive electrochemical HCHOaq detection. However, the cost and limited supply of noble-metals-based nanostructures is a major barrier to the development of disposable sensor platforms for HCHO_aq_ detection [24,25,26,27]. Among the noble-metals-based materials [24,25,28,29,30,31], Ni [32,33,34], within the highly active redox couple of Ni(OH)_2_/NiOOH, has excellent physical and chemical properties for HCHO_aq_ detection by virtue of its electro-analytical properties, e.g., low cost, rapid response, high sensitivity, and detection limits at ppb levels [27,35,36].

In this study, a portable, autonomous, IoT electrochemical sensor was developed to determine the HCHO in tap water. The sensor was constructed on a custom-designed sensor platform with IoT technology and a three-terminal electrode connector to plug in the SPE-based system for monitoring HCHO_aq_. The sensor platform consists of several electronic components, including a commercially available Photon WiFi Development Board, a miniaturized, static electrochemical cell (SEC), combined with a potentiostat, a USB port, and the SPE connector. The HCHO_aq_ was detected using an already-developed system based on Ni(OH)_2_–Ni nanowires (NWs) [36,37] and pSPEs. The proof-of-principle HCHO analysis was performed with a lab-grade potentiostat, showing high sensitivity, rapid response (1 s), and a low-level detection capability (24 ppb). Later, the Ni(OH)_2_–Ni NWs pSPEs was connected to an assembled sensor platform to test it as a lab-on-a-chip for monitoring HCHO_aq_. The measurements showed comparable values with a commercial potentiostat and the developed portable, autonomous, IoT electrochemical sensor. Furthermore, the developed sensor was also able to detect HCHO in tap water.

## 2. Materials and Methods

### 2.1. Chemicals and Materials

Anodized aluminum oxide (AAO) templates were purchased from Whatman Anodics Filter Membranes, Merck, Ljubljana Slovenia, the gold electrolyte, Ecolyt SG100, was purchased from Gramm, Heimerle + Meule GmbH, Pforzeim, Germany, and double-sided Cu-foil tape was purchased from 3M^TM^, Saint Paul, MN, USA. Formaldehyde (HCHO_aq_, 37% *w*/*v*) was purchased from Carlo Erba, Cornaredo, Italy. Nickel sulfate hexahydrate (NiSO_4_∙6H_2_O, 98+%) was purchased from Acros Organics, Geel, Belgium. Boric acid (H_3_BO_3_), sulfuric acid (H_2_SO_4_, 99.9%), sodium hydroxide pellets (NaOH), and potassium hydroxide pellets (KOH) were purchased from Sigma-Aldrich, St. Louis, MO, USA.

### 2.2. Preparation of the HCHO Detection System Based on Ni(OH)_2_–Ni NWs pSPEs

The Ni(OH)_2_–Ni NW-based receptor elements (Figure 1) were prepared electrochemically. The experiments were performed in a Teflon electrolytic cell, on a Gamry Reference 600 potentiostat at room temperature. A standard three-electrode system was used; the reference electrode (RE) was a conventional Ag/AgCl/3.5 M KCl (HANA Instruments; HI5311), and a hollow-circular platinum mesh was used as the counter electrode (CE). To prepare the Ni(OH)_2_–Ni NW-based receptor elements for HCHO_aq_ detection, two-step electrochemical synthesis routes were employed. The first step covers the amperometric deposition of free-standing Ni NWs. The Ni NWs were electrodeposited by applying a constant potential of −1.0 V for 1200 s into commercially available anodized aluminum oxide (AAO) membranes (Whatman Anopore, pore diameter ~200 nm) using 0.2 M NiSO_4_∙6H_2_O and 0.1 M H_3_BO_3_ aqueous solution. The pH of the prepared solution was adjusted to 2.0 by adding H_2_SO_4_ [38]. After the deposition, the AAO membrane was removed in 10 M NaOH solution in order to obtain a one-dimensional material, i.e., free-standing Ni NWs. Detailed information about the electrochemical deposition of Ni NWs can be found in our recent publication, i.e., Trafela et al. [36]. Furthermore, the as-prepared Ni NWs were electrochemically modified (i.e., potential cycling at a scan rate of 200 mV/s) in the 0.1 M KOH electrolyte. The second step, i.e., KOH modification, was chosen as it enables an effective transformation of the Ni NWs surface to highly active Ni(OH)_2_ for HCHO_aq_ detection [37]. For more detailed information about the KOH modification process, please refer to our publications [36,37]. The resulting homogeneous Ni(OH)_2_–Ni NWs, with an average length of 2 µm, diameter of 200 nm, and Ni(OH)_2_ surface layer thickness of 8–10 nm, were characterized by FEG-SEM, TEM, FT-IR, and XRD [36,37], and further used as a receptor element for sensor manufacturing.

The screen-printed electrodes (Figure 1) were fabricated using conductive ink printed on synthetic paper (Monotex L, 254 g/m^2^), i.e., pSPE, using a GTO EVO 570 machine (GTO, Emilia Romagna, Italy). The pSPE was designed to fit the commercial potentiostat measuring system [16]. The printing process was performed in three layers with two different printing inks, i.e., Ag ink (CRSN 2442, SunTronic Silver 280; SunChemical, Parsippany-Troy Hills, NJ, USA) and dielectric ink (CFSN6057 SunTronic Dielectric 681; SunChemical, Parsippany-Troy Hills, NJ, USA). Two layers of the conductive Ag ink were screen-printed (mesh count of 77–55 L cm^−1^) on the paper to form the electrodes. Firstly, the quasi-RE (q-RE) and the working electrode (WE) were printed and cured at 110 °C for 30 s, and, secondly, the CE was printed and cured at 90 °C for 30 s. Finally, the layer of dielectric ink was screen-printed (mesh count of 90–48 L cm^−1^) and cured at 90 °C for 30 s. After printing, the pSPEs were additionally cured at 110 °C for 150 s [16,39].

The integration of the Ni(OH)_2_–Ni NW-based receptor element onto the pSPEs (Figure 1): the as-prepared receptor elements, i.e., Ni(OH)_2_–Ni NWs, were attached to the WE (φ = 4 mm) of the pSPE using double-sided Cu-foil adhesive tape. The resulting electrode, i.e., the HCHO_aq_ detection system (Figure 1), was then used for the electrochemical detection of HCHO.

### 2.3. Electrochemical Measurements Using Commercially Available Laboratory Equipment

The electrochemical testing of the HCHO detection system (i.e., Ni(OH)_2_–Ni NWs pSPEs) was carried out via cyclic voltammetry (CV) and chronoamperometry (CA) using commercially available laboratory equipment (Figure 1). The Ni(OH)_2_–Ni NWs pSPEs were placed in a boxed DropSens connector, connected to a Gamry Reference 600 potentiostat. The sample sensing areas (i.e., Ni(OH)_2_–Ni NWs) were wetted with the 1 mL working solution containing known concentrations of HCHO (from 0 to 1 × 10^−2^ M) and 0.1 M NaOH (pH = 13.7). All the measurements were made at room temperature. The cyclic voltammograms were recorded from 0.3 V to 1.0 V vs. q-RE at a scan rate of 100 mV/s. The CA measurements were executed at constant potentials of 0.7 V vs. q-RE for 3–5 s.

### 2.4. Sensor Manufacture

The IoT electrochemical sensor for HCHO detection in alkaline media consists of two main components (i and ii, Figure 1): (i) the Ni(OH)_2_–Ni NWs pSPEs-based detection system (the preparation of the system is described in Section 2.2) placed in a connector of (ii) a sensor platform. The sensor platform was constructed on an electronic breadboard kit and consisted of a miniaturized potentiostat, a Photon WiFi Development Board (Particle lnc., San Francisco, CA, USA), and a USB port. The detection system and potentiostat together represent a static electrochemical cell (SEC) which gives 2.5 W of single-supply power as the CA measurements were performed when a constant voltage was maintained. During the measurement time, the only electrochemical method available was chronoamperometry, so the potential was set (independent variable) and the current was measured. The voltage range was set using a digital-to-analog converter (DAC) from 0 V to 1.5 V. Furthermore, the SEC was adapted to be suitable for connection to the Photon WiFi Development Board, and the IoT. The device has a powerful industrial microcomputer and a set of peripherals that provide versatility: an ARM M4 processor (120 MHz), a flash memory for storing software, a RAM for storing various data, and a Wi-Fi system (802.11 b/g/n). Furthermore, the IoT is connected to the PC via a USB, from where it receives 2.5 W of power (USB 2.0 enables a theoretical maximum of 500 mA @ 5 V output), and sends data to the terminal. To protect the electronics from mechanical and water/chemical damage, the potentiostat and Photon WiFi Development Board were enclosed in a water-resistant case. The protective case still makes it possible to plug in a USB port and a three-terminal electrode connector for the pSPEs (10 mm × 30 mm). The sensor itself is small in size (60 mm × 30 mm), providing low-power, long-range wireless data transfer. Additionally, it is not an expensive production process as it is fabricated with low-cost parts.

The electrochemical testing of the sensor connected with the Ni(OH)_2_–Ni NWs pSPEs detection system via a three-terminal electrode plug (Figure 1) was performed using CA. The measurements were carried out under a constant potential of 0.7 V vs. Ag q-RE. The working solutions of 1-µM HCHO and 0.1 M NaOH were prepared using distilled (DI) and tap water. Once the Ni(OH)_2_–Ni NWs receptor element was wetted with the electrolyte, the IoT-based sensor platform started the measurements, i.e., sets the counter of i to 1 and shows the voltage of 0.7 V and the output currents (μA). Since the IoT sensor is connected to the PC via a USB port, the continuously written data accessible via the TCP-IP server are sent to the terminal/cloud.

## 3. Results and Discussion

### 3.1. The Strategy of HCHO Electrochemical Detection with the Ni(OH)_2_–Ni-NWs pSPE-Based Detection System

The Ni(OH)_2_–Ni NWs pSPE-based detection system was installed in the three-terminal electrode plug of the box (DropSens, Oviedo, Spain) connected to a lab-grade potentiostat. The system was tested for HCHO detection by monitoring its oxidation ability in alkaline media. Figure 2 shows the representative cyclic voltammograms of the Ni(OH)_2_–Ni NWs pSPEs, investigated in 0–0.01 M HCHO alkaline electrolyte (0.1 M NaOH) at a scan rate of 100 mV/s. In the absence of HCHO (Figure 2, black-dot), one couple of well-defined redox peaks was observed at 0.65 V (A) in the anodic region and 0.5 V (B) in the cathodic region. The anodic peak current is attributed (I ≈ 1 mA) to the oxidation reaction of Ni(OH)_2_ + OH^−^ → NiOOH + e^−^ (A). Once the NiOOH is formed, it oxidizes HCHO. However, the substance that is involved in electrochemical reactions is not HCHO, as the studies are performed in an alkaline electrolyte. After the addition of HCHO to the 0.1 M NaOH electrolyte, the formaldehyde exists in the hydrated form of methylene glycol, i.e., CH_2_(OH)O^−^ [40], and is therefore oxidized at the surface of the NWs through the chemical reaction: NiOOH (formed at A) + CH_2_(OH)O^−^ → Ni(OH)_2_ (i.e., newly formed) + CH_2_(O)O^−^ [36,37,41,42,43,44]. From Figure 2 and Figure 3, it is clear that the anodic peak current (A) increases with increasing HCHO concentration from 1 µM to 50 mM, indicating the additional oxidation of the newly formed Ni(OH)_2_ to NiOOH [36,37]. Furthermore, an examination of CVs shows that the anodic peak potential (A) shifts to a more positive potential with increasing HCHO concentration. This is attributed to the IR drop arising from the high current values at the high HCHO concentrations [36,45]. In the cathodic region, the cathodic peak current (B) was observed at the potential +0.5 V, attributed to the reversible transformation, i.e., the reduction of NiOOH + H_2_O + e^−^ → Ni(OH)_2_ + OH^−^ (B). Thus, these reduction (A) and oxidation peaks (B) correspond to the reversible reaction: Ni(OH)_2_ + OH^−^ ↔ NiOOH + e^−^.

The maximum, still sensitive, HCHO concentration was found to be ≤10 mM. Above this optimal HCHO concentration, additional anodic (Figure 3C,E) and cathodic (Figure 3D) peaks were observed. The anodic peak current (C) at E ≈ 0.5 V (orange) or E ≈ 0.6 V (grey) indicates a measurable side effect, an additional reaction occurring at Ag q-RE of the pSPE [46]. The Ag q-RE can interfere with the electrochemical measurements and cause non-negligible artifacts, which are attributed to the redox reactions of Ag in HCHO alkaline media [46,47,48,49]. The additional anodic peaks (C, Figure 3) are related to the electro-oxidation of Ag to [Ag(OH)2]^−^ and finally to Ag oxides (Ag_2_O/AgO). The results indicate that the HCHO is indeed oxidized by the Ag species, and the one with a higher valence (Ag_2_O/AgO) oxidizes the formaldehyde (to final product, probably CO_2_) and is followed by the generation of low-valence Ag [41]. From Figure 3, it is clear that the reactivity of the HCHO with Ag increases with its concentration. Hence, the additional cathodic peaks (D, Figure 3) at E ≤ 0.3 V (grey curve; not visible in this potential range for orange curve) can be attributed to the reversible processes involved in the reduction of Ag_2_O/AgO to Ag. The typical behavior for Ag q-RE in alkaline media and a broader discussion of the oxidation–reduction results are mentioned elsewhere [46,47,48,49]. Furthermore, a tail of the signal current was observed at E ≈ 0.93 V (E, Figure 3). This tail could be attributed to the bubble burst (e.g., oxygen that is formed during the oxygen-evolution reaction at a high anodic potential, or CO_2_ which can be formed during the reaction between HCHO and Ag q-RE). This means that the Ni(OH)_2_–Ni NWs pSPE begins to saturate and the formed bubble hinders the HCHO oxidation [45]. Based on these findings, the target concentration range for amperometric HCHO detection was selected to be from 1 µM to 10 mM. The determination of HCHO in alkaline media using the Ni(OH)_2_–Ni NWs pSPEs-based detection system was then investigated using both laboratory equipment and the developed IoT electrochemical sensor.

### 3.2. HCHO Electrochemical Detection Using Laboratory Equipment

The amperometric HCHO_aq_ detection in alkaline media was first tested using the Ni(OH)_2_–Ni NWs pSPEs detection system connected to the laboratory potentiostat via a three-terminal electrode plug on the DropSens box (Figure 1). Figure 4a shows typical current–time plots observed for 0–10 mM HCHO in 0.1 M NaOH at an operating potential of 0.7 V. As seen from Figure 4a, a sharp rise in the current was observed in the first second, which was followed by an exponential drop until the steady state was reached. Each peak in the first second represents an increase in current due to the oxidation reaction of Ni(OH)_2_ (reactions 1 and 3), which demonstrates the rapid response and the high sensitivity of the Ni(OH)_2_–Ni NWs pSPEs. The calibration (Figure 4b) was made by plotting the amperometric response of the lab potentiostat versus the HCHO_aq_ concentrations (0.8 µM–10 mM). Two linear relations were obtained, i.e., from 0.8 to 500 µM (red line) and 0.5–10 mM (blue line). The experiments for each concentration were carried out in triplicate. Considering the limit of quantification (LOQ) for the Ni(OH)_2_–Ni NWs-based receptor element, its value was found to be 0.8 µM/24 ppb (Figure 4a, light green). In comparison with some previous works where Ni-based electrodes were used for the detection of HCHO in 0.1 M NaOH, it seems clear that Ni(OH)_2_–Ni NWs pSPE can act as a comparable electrode/catalyst in HCHO oxidation (Table 1).

Furthermore, the reusability of the Ni(OH)_2_–Ni NWs pSPEs was investigated with the CA responses in the 1 mM HCHO + 0.1 M NaOH solution three times. The results showed that the output current values are very close, with a relative standard deviation (RSD) of 0.37%. Another significant feature of the Ni(OH)_2_–Ni NWs pSPEs in a practical application is their reproducibility. We studied the CA responses of ten different Ni(OH)_2_–Ni NWs pSPEs to the addition of 1 mM HCHO to a 0.1 M NaOH solution. The ten Ni(OH)_2_–Ni NWs pSPEs were very reproducible, since the RSD of their output current responses was 0.63% [43]. Furthermore, considering the stability of the electrochemical responses of Ni(OH)_2_–Ni NW-based receptor elements, there is no requirement to repeat the conditioning step while the measurement potential is applied. High selectivity is also another important property of the proposed material. The Ni(OH)_2_–Ni NWs pSPE was tested for four potential interferences, including ethanol, 1-propanol, acetaldehyde, and hydrogen peroxide. It was shown that these related compounds do not contribute to the HCHO response in alkaline electrolytes [43].

### 3.3. HCHO Electrochemical Detection Assisted by an IoT Electrochemical Sensor (Lab-on-a-Chip)

The detection of amperometric HCHO_aq_ in alkaline media was also tested using a portable IoT electrochemical sensor (Figure 1). Figure 5a represents the individual parts of the fabricated IoT electrochemical sensor consisting of the integrated Ni(OH)_2_–Ni NWs pSPEs, the miniaturized potentiostat (i.e., the static electrochemical cell, SEC), the IoT-enabled platform (i.e., commercially available Photon WiFi Development Board), and a USB port. A sensor enables all the necessary operations, including the digitization of signals, voltage regulation between the q-RE and the WE of the SPEs, recording data at the required time, instantly displaying the data on the PC terminal, rapid communication via LED (both wired and wireless), and TCP-IP access to various settings.

Figure 5b depicts the amperometric response of the custom-designed IoT sensor to the dripping of 0.1 M NaOH electrolyte (green) or 1 μM HCHO + 0.1 KOHM NaOH electrolyte (red) onto the WE of the SPE (i.e., Ni(OH)_2_–Ni NWs). The amperometric measurements were performed at a constant applied potential of 0.7 V vs. Ag q-RE. In the absence of HCHO (green), the output current reaches the maximum values in the first second, indicating a rapid oxidation reaction of Ni(OH)_2_ to NiOOH (reaction 1), followed by a current-output drop due to the analyte diffusion restriction. When a 1 μM concentration of HCHO was added to the 0.1 M NaOH electrolyte (red), the maximum current output (0.34 mA) was also observed in the first second, meaning that the HCHO oxidation started directly after the formation of the first NiOOH species (reaction 3). Hence, the response time required for the sensor platform to reach its maximum current was 1 s. This indicates that the sensitivity of the proposed sensor for HCHO_aq_ detection depends on the efficiency of the current output at the start of the measurement. Furthermore, to determine the accuracy of the amperometric measurements observed with the custom-designed sensor, the current–time plots were compared with the amperometric measurements observed using the laboratory potentiostat (Figure 5b, black and grey). The current outputs in the first second are comparable for the commercial potentiostat and the developed sensor, indicating that the fabricated sensor platform does not affect the sensitivity of the HCHO_aq_ detection. However, after that point, the curves differ as the settling time of the lab-grade potentiostat is shorter than that of the fabricated IoT electrochemical sensor. Hence, the obtained data demonstrate that the developed IoT electrochemical sensor would suit applications for rapid (1 s) and low-cost (i.e., considerably cheaper than any lab-grade potentiostat) monitoring of HCHO in water.

## 4. Conclusions

In summary, we have constructed a portable, autonomous, IoT electrochemical sensor that detects HCHO_aq_ in distilled or tap water. The sensor comprises assembled electronics consisting of a Photon “Internet of Things (IoT)” development kit, a USB port for charging, and a miniaturized potentiostat, all enclosed in a small, protective case (size 58 mm × 33 mm). The external part of the sensor can connect to the HCHO detection system via a three-terminal electrode connector. The HCHO detection system based on custom-made Ni(OH)_2_–Ni NWs pSPEs allowed us to rapidly (1 s) detect HCHO_aq_ in alkaline media. The detection system can detect concentrations of HCHO at 0.8 µM/24 ppb, a linear response over two wide ranges of HCHO concentrations (0.8 × 10^−6^–5 × 10^−4^ M and 5 × 10^−4^–1 × 10^−2^ M). It also assembles the basic requirements for real-time applications, including low cost, simplicity, selectivity, and sensitivity, together with reusability and reproducibility. Thus, this study demonstrated that the developed IoT electrochemical sensor, which also allows for the easy replacement of an external detection system, can be used as an effective, portable, inexpensive device for testing HCHO and other organic/inorganic pollutants in real-life conditions.

## Figures and Tables

**Figure 1 sensors-23-04676-f001:**
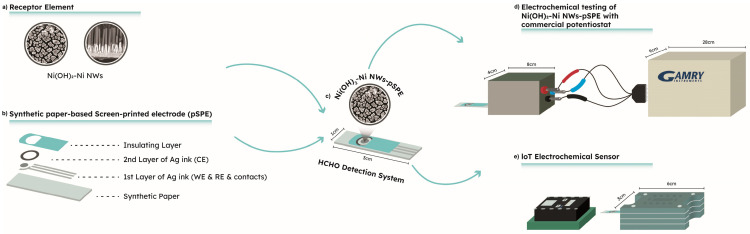
Schematic of the IoT electrochemical sensor’s fabrication process: (**a**) FEG-SEM images of the Ni(OH)_2_–Ni NW-based receptor element: left—top view and right—side view (the original FEG-SEM images are available in our recent publication, i.e., Trafela et al. [36]). (**b**) The synthetic-paper-based screen-printed electrodes (pSPEs) with individual (ink) layers, and (**c**) the fabricated HCHO_aq_ detection system with its dimensions. Design and dimensions of (**d**) laboratory equipment (Gamry) and (**e**) IoT fabricated electrochemical sensor for HCHO testing.

**Figure 2 sensors-23-04676-f002:**
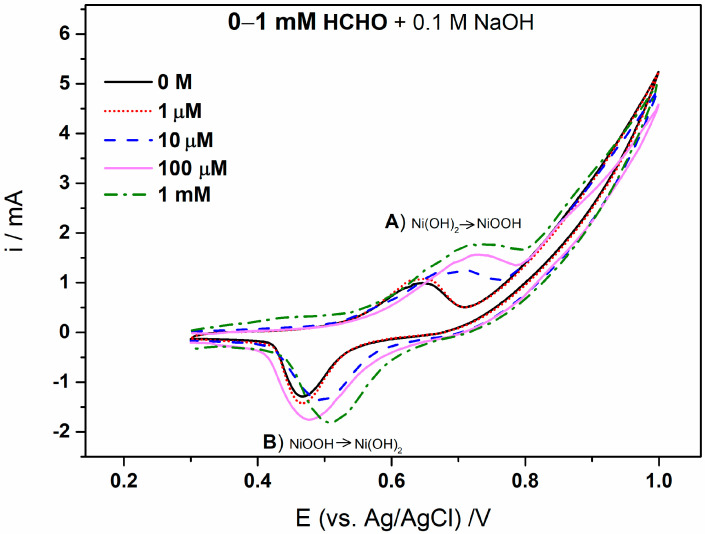
CVs of the Ni(OH)_2_–Ni NWs pSPEs obtained at different HCHO concentrations (from 0 to 1 mM) in 0.1 M NaOH solution, observed at a scan rate of 100 mV/s.

**Figure 3 sensors-23-04676-f003:**
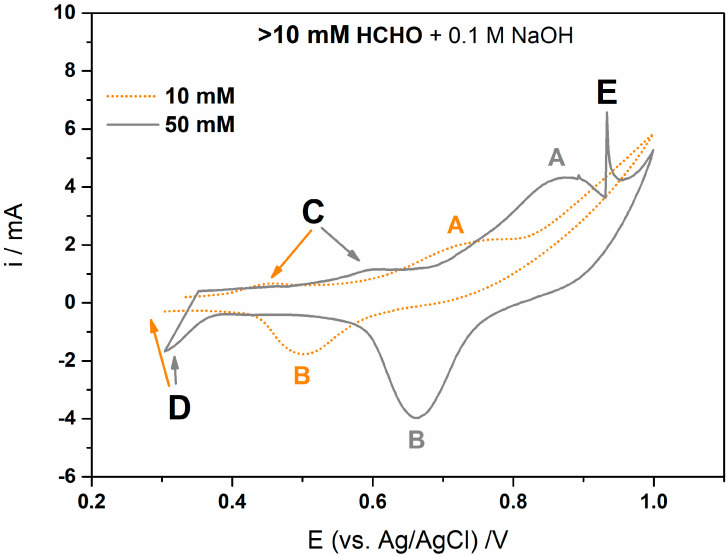
CVs of the Ni(OH)_2_–Ni NWs pSPEs obtained at different HCHO concentrations: 10 (orange) and 50 (grey) mM in a 0.1 M NaOH solution, observed at a scan rate of 100 mV/s.

**Figure 4 sensors-23-04676-f004:**
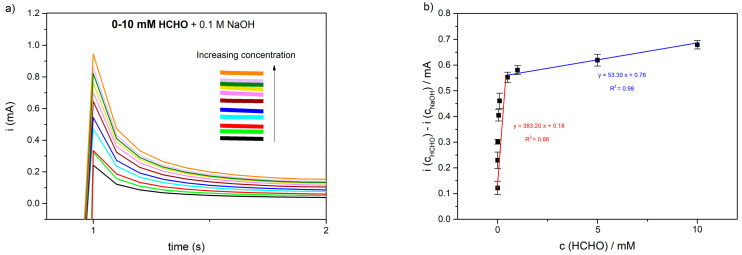
(**a**) CA responses of the Ni(OH)_2_–Ni NWs pSPEs observed with lab-grade potentiostat and HCHO concentrations: 0.0—(black), 0.8—(light green), 1—(red), 5—(cyan), 10—(blue), 50—(brown), 100—(pink), 50—(yellow) µM, 1—(green), 5—(purple), and 1—(orange) mM in 0.1 M NaOH, observed at an applied potential of 0.7 V; (**b**) calibration plot of HCHO detection. Black square dots are oxidation peak values for HCHO concentrations, obtained in the first second of CA measurement (Figure 4a). Error bars are the standard deviation from the mean of three replicates.

**Figure 5 sensors-23-04676-f005:**
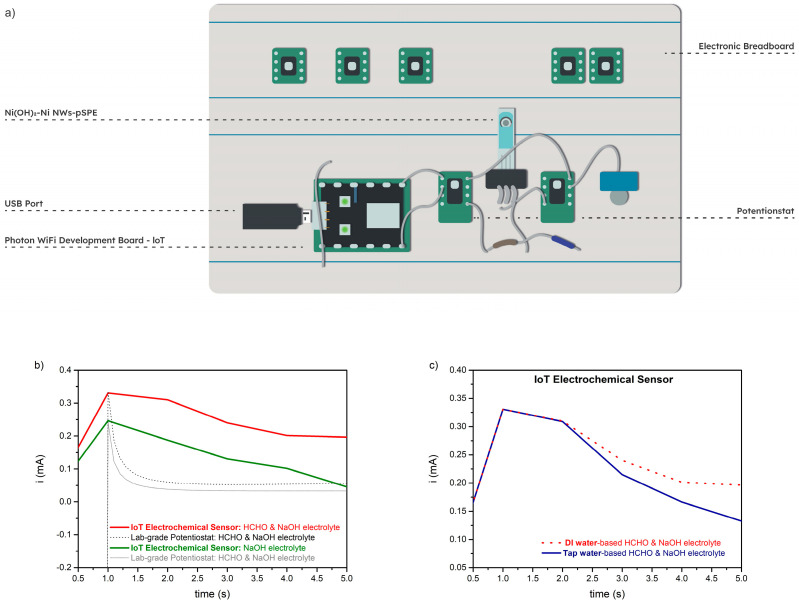
(**a**) Schematic of the IoT electrochemical sensor’s individual parts with an integrated Ni(OH)_2_–Ni NWs pSPEs-based detection system; (**b**) amperometric responses of custom-made IoT electrochemical sensor to injection of 0.1 M NaOH electrolyte (green) and 1 µM HCHO + 0.1 M NaOH electrolyte (red). Amperometric responses of lab-grade potentiostat observed in 0.1 M NaOH electrolyte (grey) and 1 µM HCHO + 0.1 M NaOH electrolyte (black); (**c**) amperometric responses of custom-made IoT electrochemical sensor to the injection of 1 µM HCHO + 0.1 M NaOH prepared in DI—(red) and tap-water-based (blue) electrolytes. All the curves were recorded at an applied potential of 0.7 V.

**Table 1 sensors-23-04676-t001:** Comparison of the Ni(OH)_2_–Ni NWs pSPE for HCHO oxidation with some of the previously reported Ni-based electrodes.

Ni-Based Electrodes	Electrolyte	Operating Potential [V]	LOQ [mM]	Saturation Limit of HCHO [mM]	Ref.
Ni(OH)_2_/POT (TX-100)/MCNTPE	0.1 M NaOH	0.7	4.9	48	[33]
Ni/P-nanozeolite-modified electrode	0.1 M NaOH	0.58	5.8	>11.5	[27]
Ni/P(NMA)MCPE	0.1 M NaOH	0.74	8	70	[34]
Ni-CHIT/CPE	0.1 M NaOH	0.56	3	90	[32]
Ni(OH)_2_–Ni NWs pSPE	0.1 M NaOH	0.7	0.8	50	Our work

## Data Availability

Availability Statements are available in section “MDPI Research Data Policies” at https://www.mdpi.com/ethics (accessed on 15 March 2023).

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
