# Peer review of "IoT Electrochemical Sensor with Integrated Ni(OH)2–Ni Nanowires for Detecting Formaldehyde in Tap Water"

_sensors, 2023, doi:10.3390/s23104676_

Round 1

Reviewer 1 Report

The manuscript is about formaldehyde detection using Ni-Ni(OH)2 modified screen printed electrode. Although the detection principle is well known but the approach is very actual now for IoT systems. The paper has really good quality and a couple of minor changes are needed.

1. English needs slight revision. Especially, "sensibility" needs to be replaced with "sensitivity".

2. Fig. 3: Is peak E reproducible? If it is due to oxygen bublle it should be a bit different in each scan or with each repetition.

Author Response

A point-by-point responce to the Reviewer #1 is provided in the attachment.

Reviewer 2 Report

Dear Trafela et. to the. your manuscript “IoT Electrochemical Sensor with Integrated Ni(OH)2-Ni Nanowires for Detecting Formaldehyde in Tap Water”, is quite interesting but I have the following observations:

Comment #01: In line 95, they mention an Ag/AgCl reference electrode, but what is the concentration of KCl, 0.1, 3, 3.5 M or saturated?

Comment #02: Line 98 talks about the electrodeposition of Ni NWs, but what was the potential that was applied, pH of the synthesis, supporting electrolyte and its concentration?

Comment #03: Between lines 100-104 it mentions the synthesis of Ni(OH)2, but I find the text confusing and I don't understand the synthesis?

Comment #04: In lines 128 and 129 you mention the supporting electrolyte, but why did you choose it and how did you choose its concentration?

Comment #05: In line 167 it shows the sensor schematic, but it is not clear how I synthesize the Ni(OH)2-Ni NW.

Comment #06: In Figure 2, cyclic voltammetry is performed for the detection of HCHO, where in all of them it shows that the material oxidizes HCHO, but at the 1 mM concentration there is also a reduction of HCHO, why does this occur, what do you attribute it to? It is not clear to me and it causes a lot of noise

Comment #07: In figure 2, when 100 µM HCHO is added, the potential for oxygen evolution decreases, why is this? Why do the other concentrations present the same OER curves?

Comment #08: Figure 3 causes more questions than certainties, explain it better in the text, in addition peak C, justifies it with the reaction of Ag to [Ag(OH)2)]-, this indicates that the electrode was more manufactured , and because this peak increases with the concentration of HCHO, if it does not interact with Ag as described? Does the E noise justify it with the appearance of an oxygen bubble? Does this always occur or only 1 voltammetry record? .

Comment #09: The material must be characterized by SEM-ESD, AFM, XPS, XRD, etc., you must be certain of the material, structure and size of these. Also, corroborate that it is a homogeneous synthesis.

Comment #10: The ESCA of this material should be calculated, it would be a contribution to your manuscript.

Comment #11: You should compare your sensor to others described in the literature, as I don't see any comparison to what is already reported in the literature.

Author Response

A point-by-point response to the Reviewer #2 is provided in the attachment. 

Round 2

Reviewer 2 Report

Dear Tafarela:

I am satisfied with your modifications, I recommend that the manuscript be accepted.